# A Case of COVID-19 Pregnancy Complicated with Hydrops Fetalis and Intrauterine Death

**DOI:** 10.3390/medicina57070667

**Published:** 2021-06-28

**Authors:** Daniela Eugenia Popescu, Andreea Cioca, Cezara Muresan, Dan Navolan, Arina Gui, Ovidiu Pop, Tamara Marcovici, Constantin Ilie, Marius Craina, Marioara Boia

**Affiliations:** 1Department of Neonatology, “Victor Babeş” University of Medicine and Pharmacy, Eftimie Murgu Sq. No.2, 300041 Timişoara, Romania; popescu.daniela@umft.ro (D.E.P.); marianaboia@yahoo.com (M.B.); 2Department of Neonatology, Premiere Hospital, Regina Maria Health Network, Calea Aradului, No.113, 300645 Timişoara, Romania; 3Department of Pathology, Premiere Hospital, Regina Maria Health Network, Calea Aradului, No.113, 300645 Timişoara, Romania; 4Department of Obstetrics and Gynecology, Premiere Hospital, Regina Maria Health Network, Calea Aradului, No.113, 300645 Timişoara, Romania; muresan.maria@umft.ro (C.M.); arina_vra@yahoo.com (A.G.); 5Department of Obstetrics and Gynecology, “Victor Babeş” University of Medicine and Pharmacy, Eftimie Murgu Sq. No.2, 300041 Timişoara, Romania; navolan.dan@umft.ro (D.N.); craina.marius@umft.ro (M.C.); 6Department of Morphological Sciences, University of Oradea, Universitatii Street, No.1, 410087 Oradea, Romania; drovipop@gmail.com; 7Department of Pediatrics, First Pediatric Clinic, “Victor Babeş” University of Medicine and Pharmacy, Eftimie Murgu Sq. No.2, 300041 Timişoara, Romania; marcovici.tamara@umft.ro; 8Department of Neonatology, Regina Maria Health Network, Aristide Demetriade Street, No.1, 300088 Timisoara, Romania; constantinilie@umft.ro

**Keywords:** COVID-19, coronavirus, SARS-CoV-2, pregnancy, vertical transmission, fetal death, hydrops fetalis, placenta

## Abstract

Coronavirus disease 2019 (COVID-19) has rapidly evolved into a worldwide pandemic causing a serious global public health problem. The risk of vertical transmission of SARS-CoV-2 is still debated, and the consequences of this virus on pregnant women and their fetuses remain unknown. We report a case of pregnancy complicated with hydrops fetalis that developed 7 weeks after recovery from a mild SARS-CoV-2 infection, leading to intrauterine death of the foetus. Evidence of SARS-CoV-2 placentitis was demonstrated by the presence of viral particles in the placenta identified by immunohistochemistry. As we excluded all possible etiological factors for non-immunologic hydrops fetalis, we believe that the fetal consequences of our case are related to vertical transmission of SARS-CoV-2 virus. To the best of our knowledge, this is the second reported case in the literature of COVID-19 infection complicated with hydrops fetalis and intrauterine fetal demise.

## 1. Introduction

Coronavirus disease 2019 (COVID-19) caused by severe acute respiratory syndrome coronavirus 2 (SARS-CoV-2) has rapidly evolved into a worldwide pandemic causing a serious global public health problem. Previous experience of pregnant women infected with other types of coronaviruses such as severe acute respiratory syndrome (SARS) and Middle East respiratory syndrome (MERS) suggests that pregnant women might be more vulnerable to severe SARS-CoV-2 infection than the general population [1,2]. 

The impact of SARS-CoV-2 infection in pregnant women is still debated, and new information is constantly being revealed. While there is no clear evidence of adverse effects on pregnancy in the first trimester, an increased prevalence of preterm labor and delivery was noticed in women infected with SARS-CoV-2 during the third trimester of pregnancy [3,4]. There were also other adverse outcomes described in pregnant women infected with COVID-19 such as premature ruptures of membranes, intrauterine fetal distress, intrauterine growth restriction, low birth weight, in utero fetal death, or premature neonatal death [2,4]. Moreover, some multistate surveillance studies performed in the USA have shown that the stillbirth rates increased from <1% (before the pandemic) to 2.2–3% [5,6,7]. Although these data are alarming, the clinical outcomes of COVID-19 on pregnant woman and their fetuses are not fully documented, as many countries are still in the grips of the pandemic and our knowledge about this disease is based on individual case reports or limited cohorts. To establish the true effect of COVID-19 on pregnancy, complete data from larger studies are required. 

In this paper, we report a case of pregnancy complicated with hydrops fetalis that developed 7 weeks after recovery from SARS-CoV-2 infection leading to intrauterine fetal demise in association with documented placental SARS-CoV-2 infection.

## 2. Case Report

A 27-year-old primigravida woman presented at 18 weeks’ gestation with fever of 38.2 °C, taste and smell loss, dry cough, and fatigue. A transcriptase polymerase chain reaction (RT-PCR) of nasopharyngeal swab was positive for SARS-CoV-2. C and D vitamins were administered, along with Amoxicillin, Acetaminophen, and Enoxaparin for 14 days, then the patient was discharged home as her illness did not require hospitalization.

Consanguinity was denied and her past medical history was unremarkable. TORCH screen was negative, and thrombophilia tests were also negative. 

A routine ultrasound examination (US) performed at 25 weeks’ gestation revealed hydrops fetalis with skin edema and severe thoracic and abdominal effusion (Figure 1). As she was Rhesus (RhD) negative and her husband was positive, with negative anti-RH antibodies, a dose of Anti-D immunoglobulin was administrated. No anemia was detected during pregnancy. A thoracocentesis and amniocentesis were performed. Cytogenetic analysis revealed a normal female karyotype. Because the evolution did not improve, another thoracocentesis and a percutaneous in utero thoracoamniotic shunt was placed in the left hemithorax at 26 weeks’ gestation (Figure 2). However, the evolution remained stationary, and at 28 weeks, the obstetric US confirmed fetal death. 

A stillborn fetus was delivered vaginally after 9 h of labor. The autopsy revealed a female fetus, appropriate for gestational age (28 weeks’ gestation), with systemic edema (Figure 3). There was no evidence of external or internal malformations. The placenta weighed 751 g (>90th percentiles for 28 weeks’ gestation) and presented a marginal cord insertion at 1 cm from the disc edge.

Microscopic evaluation of the internal organs depicted a systemic thrombosis, with old and recent thrombi in the small and medium vessels. There were groups of villi with stromal hemorrhage and a mild acute inflammatory infiltrate was found in subchorionic space consistent with acute subchorionitis (maternal response stage 1, grade 1 with no fetal response), (Figure 4A). Recent thrombi were identified in fetal circulation including umbilical vein (at the fetal edge), chorionic vessels, and stem villi vessels (Figure 4B). In addition, recent intervillous thrombi, perivillous fibrin deposition, and mixed inflammatory infiltrates composed of neutrophils and few monocytes were noted in the intervillous space (Figure 4C). Gram and periodic acid–Schiff staining of the placenta and culture did not demonstrate any bacterial or fungal infections. Immunohistochemistry with SARS-CoV-2 nucleocapsid protein showed strong positivity of the trophoblast and fetal villous macrophages (Figure 4D). Fetal skin was sampled for the molecular karyotype (SNP array, Cytoscan 750KAffimetrix), which showed a normal female karyotype, arr(1-22,X)x2.

## 3. Discussion

Vertical transmission of SARS-CoV-2 is a rare event, with an estimated incidence of 1–3% of cases or even lower as reported by other authors [8,9]. Although, the mechanism of vertical transmission remains controversial, detection of SARS-CoV-2 in placental or membrane tissues by immunohistochemistry or molecular methods supports a scenario of vertical transmission of the virus [8,10]. A possible pathway for viral entrance is through the angiotensin-converting enzyme 2 (ACE2), a surface sensitive cell receptor for SARS-CoV-2, that showed an aberrant expression in human placentas. Another possible explanation for intrauterine SARS-CoV-2 infection is via maternal immune cells or, less commonly, during vaginal delivery [11].

Placental pathology findings from pregnant women with COVID-19 were previously described by many authors (Table 1), [12,13,14,15,16,17,18,19,20]. Poisson and Pierone identified placental lesions such as extensive fetal vascular malperfusion in a case of fetal demise from a woman with SARS-CoV-2 infection without any associated disease [12]. In a series of 20 cases, Baergen and Heller showed that fetal vascular malperfusion was the most common lesion in their cohort [13]. Menter et al. described features of villitis and malperfusion in a series of five cases and suggested that these histopathological changes may be related to an altered coagulative or microangiopathic state induced by COVID-19 [14]. In a recent study, inflammatory and thrombo-hemorrhagic alterations were the most frequent pathological changes in a series of ten placentas from women infected with COVID-19 [15]. Ferraiolo et al. identified placental alterations, and Birindwa et al. reported a case of a woman infected with SARS-CoV-2 that gave birth to a newborn also infected with COVID-19. The baby died 5 days after delivery and histopathological examination revealed thrombotic vasculopathy of both placenta and umbilical cord [16]. In line with these reports, we found recent and organized thrombi in fetal circulation including the umbilical vein, chorionic and stem villi vessels but also in the small and medium vessels of the fetal internal organs. We also identified a mild acute inflammatory infiltrate in subchorionic space consistent with acute subchorionitis and a mixed intervillitis. In a series of five cases of fetal demise in women with mild or moderate forms of COVID-19 infection, Richtmann et al. reported acute chorioamnionitis in five cases and mixed intervillitis/villitis in two of them [17]. Baud et al. described a mixed inflammatory infiltrate in the subchorionic space and intervillous fibrin deposition in a case of miscarriage during the second trimester in a pregnant woman with COVID-19 [18]. It is well known that COVID-19 is associated with a thrombo-inflammatory state, and thromboses in human placenta were already documented [15,16,19]. These observations suggest that fetal vascular thrombosis and inflammatory changes may represent a histological marker of COVID-19 placental infection. Moreover, these data are supported by the presence of viral particles in the placenta identified by immunohistochemistry [8,10,11]. In the present case, strong positive staining for SARS-CoV-2 was noted in both trophoblast and fetal villous macrophages. We believe that in our case, the placental inflammation was linked to the viral infection; thus, these findings represent a SARS-CoV-2 placentitis.

A causal relation between COVID-19 and non-immune hydrops fetalis has not been yet demonstrated. However, a recent publication described a transitory fetal skin edema associated with polyhydramnios developed after recovery from a mild SARS-CoV-2 infection of the mother. These fetal changes presented a spontaneous resolution in utero, and no abnormalities were found in the newborn [21]. Garcia-Manau et al. reported two cases of fetal transient skin edema in pregnant women with COVID-19 in their second trimester of pregnancy. In these cases, the fetal skin edema appeared when the mothers were positive for COVID-19 infection and resolved when maternal SARS-COV-2 RT-PCR test results became negative [22]. Shende et al. described a case of first trimester asymptomatic SARS-CoV-2 infection complicated with hydrops fetalis and intrauterine fetal demise. In their case, viral RNA was identified in the amniotic fluid, and the S proteins were detected in the fetal membranes 5 weeks after the mother recovered from SARS-CoV-2 infection, concluding that hydrops fetalis and intrauterine fetal demise were caused by the congenital transmission of COVID-19 [23]. Similarly with their case, we found evidence of SARS-CoV-2 placentitis demonstrated by the presence of viral particles in the placenta identified by immunohistochemistry. As we excluded all possible etiological factors for non-immunologic hydrops fetalis, we believe that the fetal consequences of our case are related to vertical transmission of the COVID-19 virus.

To conclude, we reported a case with documented placental SARS-CoV-2 infection associated with fetal vascular thrombosis. To the best of our knowledge, this is the second reported case in literature of COVID-19 infection complicated with hydrops fetalis and intrauterine fetal demise. More data on pregnant women infected with COVID 19 and their fetuses are needed to create guidelines for clinical practice in order to prevent potential negative outcomes and fetal complications. Until more definitive answers are available, increased surveillance of pregnant women and their fetuses is needed.

## Figures and Tables

**Figure 1 medicina-57-00667-f001:**
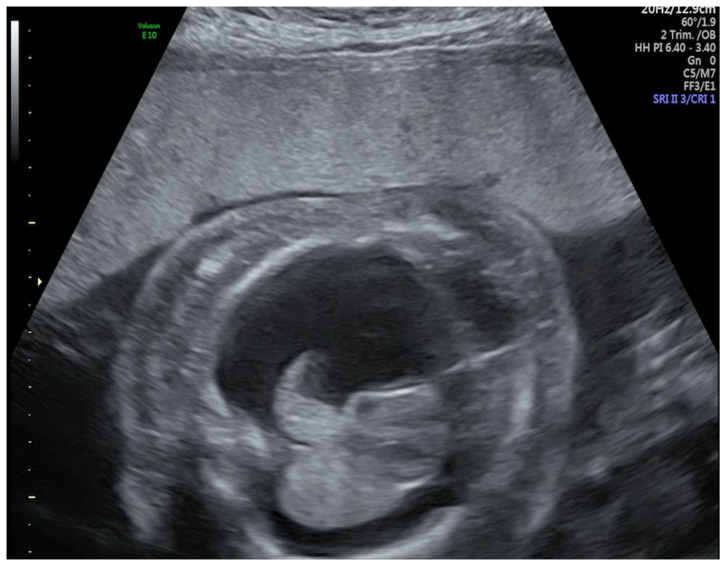
Transverse abdominal section of the thorax showing skin edema and thoracic effusion.

**Figure 2 medicina-57-00667-f002:**
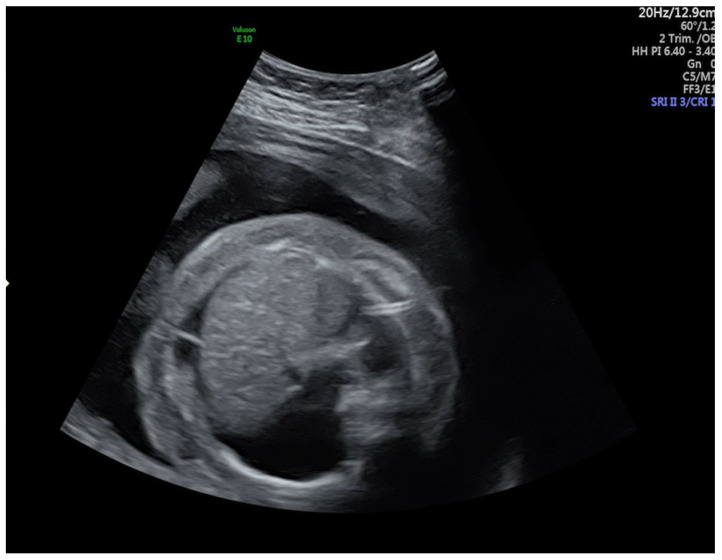
Transverse abdominal section revealing ascites and the presence of percutaneous thoracoamniotic shunt.

**Figure 3 medicina-57-00667-f003:**
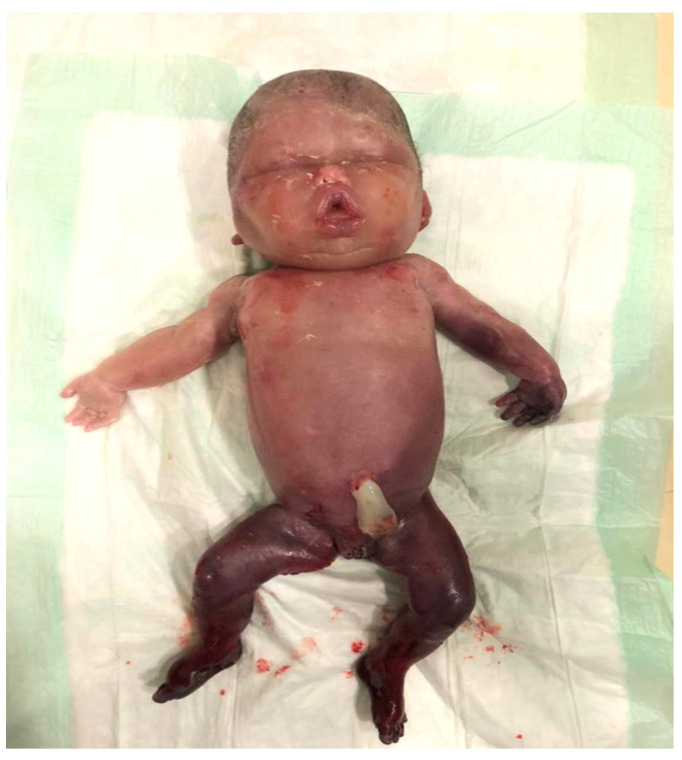
Macroscopy of the fetus showing generalized massive edema.

**Figure 4 medicina-57-00667-f004:**
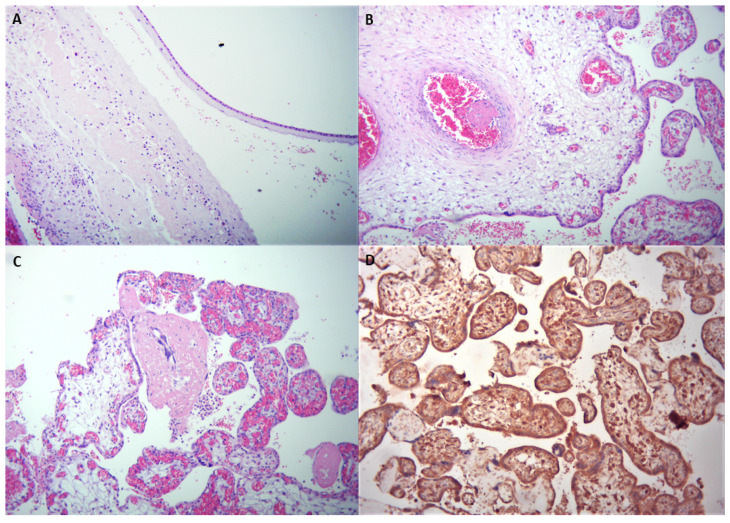
(**A**) Microscopy of the membranes showing a mild acute inflammatory infiltrate in the subchorionic space consistent with acute subchorionitis (H&E staining, 10×). (**B**). Recent thrombus attached to the endothelium in a stem villi vessel (H&E staining, 10×). (**C**). Placenta section revealing perivillous fibrin deposition and mild inflammatory infiltrate (H&E staining, 10×). (**D**). Immunohistochemistry for SARS-CoV-2 protein showing strong positivity of the trophoblast and fetal villous macrophages, 10×.

**Table 1 medicina-57-00667-t001:** Placental pathology in COVID-19 positive mothers.

Study	GA	Histopathological Findings	
		FVM	Other Findings
Poisson et al. [12]	35 weeks	Thrombosis, Avascular villi	Acute chorionitis, Maternal vascular malperfusion
Baergen et al. [13]	33 weeks–40 weeks	Thrombosis, Fibrin deposition, Karyorrhexis, Chorangiosis	Maternal vascular malperfusion Intervillous thrombus, Focal increase in fibrin
Menter et al. [14]	39 weeks–41 weeks	Thrombosis, Avascular villi, Chorangiosis, Delayed villous maturation	Chorioamnionitis, Chronic vilittis, Subchorionitis, Chronic deciduitis, Maternal vascular malperfusion
Bertero et al. [15]	32 weeks–40 weeks	Thrombosis, Chorangiosis, Accelerated maturation, Fibrin deposition, Avascular villi	Chronic villitis, Intervillous hematoma, Maternal vascular malperfusion
Birindwa et al. [16]	34 weeks	Thrombosis, Chorangiosis	
Richtmann et al. [17]	21 weeks–38 weeks	Fibrin deposition	Chorioamnionitis, Chronic villitis, Acute deciduitis
Baud et al. [18]	19 weeks		Subchorionitis, Funisitis
Shanes et al. [19]	33 weeks–40 weeks	Fetal vessel mural fibrin, Avascular villi, Delayed villous maturation, Chorangiosis	Maternal vascular malperfusion
Ferraiolo et al. [20]		Microchorangiosis, Fibrin deposition	Subchorionitis, Intervillous hemorrhages

GA, weeks of gestation; FVM, fetal vascular malperfusion.

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
