# Peer review of "A Case of COVID-19 Pregnancy Complicated with Hydrops Fetalis and Intrauterine Death"

_medicina, 2021, doi:10.3390/medicina57070667_

Round 1

Reviewer 1 Report

This is an interesting case report addressing the issue of vertical transmission in patient with SARS Cov2 infection and possible association of covid infection and serious adverse pregnancy outcome. There are still limited data on this aspect and this case report brings valuable information in understanding the real implication of COVID-19 infection during pregnancy.

  1. As the main findings in histological examination were thromboses into the placenta and fetal microvascular thrombosis, is there any benefit to continue the antithrombotic therapy with low molecular weight heparin for longer period or eventually until the end of pregnancy? For patients confirmed positive for SARS Cov 2 at the beginning a pregnancy would you consider starting Aspirin therapy, considering that covid infection also increases the risk of developing preeclampsia?
  2. Is there any evidence that invasive procedures (amniocentesis or thoracocentesis) performed in pregnant women confirmed positive for covid might increase the risk of vertical transmission? Please address this issue.
  3. Please present in a table all cases reported in the literature regarding placental analysis in COVID cases

Reviewer 2 Report

In this case report the authors present an interesting case of a pregnant patient positive for SARS-CoV-2 infection and a fatal pregnancy complication- few weeks apart the fetus developed hydrops and intrauterine demise was subsequently identified. Most common causes of nonimmune fetal hydrops were excluded and the remaining etiological cause was related to SARS-CoV-2 infection. In view of the presenting case, I would like to address to the authors the following questions:

  1. Did the authors test for intrauterine infections in the amniotic fluid sample: CMV, toxoplasmosis, syphilis?

  1. What is the supposed mechanism for fetal hydrops or transitory edema in relation to covid infection?

  1. Why did the authors not investigate the presence of the virus into the amniotic fluid or into the various fetal organs (thoracocentesis, pulmonary)? This would have been an important evidence of intrauterine vertical transmission.

  1. There has been a long period between first detection of maternal systemic viremia and placental viral detection and fetal complications onset. Is there a recommended monitoring plan for pregnant women positive for covid infections? Would you recommend more intense monitoring several weeks post-infection? Please comment on the current guidelines on pregnancy monitoring with SARS Cov2 infection, taking into account the higher risk of complications, especially the risk of stillbirth.
